# GAUSSIAN DIFFERENTIALLY PRIVATE HUMAN FACES UNDER A FACE RADIAL CURVE REPRESENTATION

**Carlos Soto**
Department of Mathematics and Statistics
University of Massachusetts Amherst
Amherst, MA 01003, USA
`carlossoto@umass.edu`

**Matthew Reimherr**
Department of Statistics
Pennsylvania State University
University Park, PA 16802, USA
`mlr36@psu.edu`

**Aleksandra Slavkovic**
Department of Statistics
Pennsylvania State University
University Park, PA 16802, USA
`abs12@psu.edu`

**Mark Shriver**
Department of Anthropology
Pennsylvania State University
University Park, PA 16802, USA
`mds17@psu.edu`

## ABSTRACT

In this paper we consider the problem of releasing a Gaussian Differentially Private (GDP) 3D human face. The human face is a complex structure with many features and inherently tied to one's identity. Protecting this data, in a formally private way, is important yet challenging given the dimensionality of the problem. We extend approximate DP techniques for functional data to the GDP framework. We further propose a novel representation, face radial curves, of a 3D face as a set of functions and then utilize our proposed GDP functional data mechanism. To preserve the shape of the face while injecting noise we rely on tools from shape analysis for our novel representation of the face. We show that our method preserves the shape of the average face and injects less noise than traditional methods for the same privacy budget. Our mechanism consists of two primary components, the first is generally applicable to function value summaries (as are commonly found in nonparametric statistics or functional data analysis) while the second is general to disk-like surfaces and hence more applicable than just to human faces.

## 1 INTRODUCTION

In statistical analyses, data and parameters appear in varying degrees of complexity, from simpler forms such as scalars and vectors to more complex such as spherical or hyperbolic, for instance. The structural constraints inherent to data need to be respected throughout any analysis as has been shown in the "intrinsic statistics" frameworks (Pennec, 2006; Bhattacharya & Patrangenaru, 2003) for accurate estimation and to preserve said structure. Complex data structures tend to come hand in hand with complex statistical computations and hence the techniques for handling such data have not been widely studied outside of specific scenarios. Further, the sheer amount of data that is captured from individuals has increased significantly, and of course, this produces a growing concern for one's "privacy". In this paper, to support broader sharing of confidential data, we propose releasing a *Gaussian Differentially Private*, GDP, average 3D human face.

**Motivation and Related Literature:** Data that live in nonlinear spaces can be challenging to work within the DP framework, as shown in the context of manifolds in Reimherr et al. (2021); Soto et al. (2022); Utpala et al. (2022), private Riemannian optimization in Han et al. (2022), and Gaussian DP on manifolds Jiang et al. (2023). Preservation of structure has also been considered in the context of private covariance estimation for linear regression in Sheffet (2019) and private principal component analysis in Chaudhuri et al. (2013) which is connected to the Stiefel manifold. For privacy, our proposed "face radial curve" representation of a human face are functions extracted from a disk and hence lends itself to be examined under the lens of private functional data analysis (FDA) which has been considered inWasserman & Zhou (2010) and Mirshani et al. (2017), and references therein.

With our faces constantly being captured (e.g., at a grocery store self-checkout), one could question the privacy protections in place of the collected data. Further, there is a vast amount of literature on identification of individuals from facial data in the area of "face recognition," see, for instance, the expansive literature review in Kortli et al. (2020). Also, perhaps quite surprisingly, Venkatesaramani et al. (2021) and Klimentidis & Shriver (2009) showed that one could identify genomic information from a person's 2D face image; the former study further showed that adding noise to said 2D images can help protect this re-identification. For 2D images, two common practices to attain privacy of faces are blurring and pixelation (Li & Choi, 2021; Vishwamitra et al., 2017). A major goal in these previous works is privacy in form of *anonymity* which is usually measured by re identification of individuals; our work, however, defines privacy by satisfying the conditions of *differential privacy*. This distinction is necessary as the former, generally, considers successful privacy by lack of re identification while preserving utility while for the latter differential privacy is a property of our mechanism. We do not work with 2D images here, however, the need for privacy is still present. The need for privacy for 3D faces is similar to that in 2D images. Anthropological studies are often interested in identification of DNA using labeled 3D faces (Sero et al., 2019), the connections between DNA genotype and the associated facial phenotype (White et al., 2021; Naqvi et al., 2021; Weinberg et al., 2019), and average faces for demographics such as age and race classification (Tokola et al., 2015). In such studies, one might want to release the data, or may even be required to, so offering provable confidentiality guarantees for people in the studies becomes an important task.

To generate our representation we rely on tools from *shape analysis*. The earlier forms of shape analysis, such as Kendall's shape space (Kendall, 1984), are limited to only representing a shape as a finite point cloud. The field has expanded since to consider more complex data structures such as continuous curves, both planar and space, as in Trouvé & Younes (2005); Klassen et al. (2004); Srivastava et al. (2010), and surfaces as in Jermyn et al. (2017); Su et al. (2020). Human faces have been considered in this space in a similar, yet subtly different, manner such as in Samir et al. (2006); Drira et al. (2010) in which faces are represented as a set of curves "facial curves" and "radial curves", respectively. These methods represent a face with curves that are generated independently of each other while we use an entire disk-parameterization to capture features across all faces simultaneously.

**Main Contributions:** We develop a novel representation for a collection of 3D faces via a set of curves which we extract from disk parameterizations and refer to these as *face radial curves*. We construct the face radial curves using tools from statistical shape analysis in the interest of preserving the *shape* of an average face during the data sanitization process. Further, we extend existing approximate DP FDA tools into the Gaussian DP framework (Dong et al., 2019), a recent notion of privacy with attractive properties including "tight" composition. Under our FDA Gaussian DP mechanism, we generate a private average face of a collection of faces under our representation. While we use our mechanism and representation for faces to address the inherent data privacy concerns, this same methodology can be applied to any surfaces diffeomorphic to a unit disk such as terrain models and additionally any settings where one wants private functional statistical summaries.

## 2    NOTATION AND BACKGROUND ON FACE REPRESENTATION

We require a disk-parameterized representation of each face from which we extract a set of functions. We parameterize each face independently but "align" and "register" each face to a template. Here, we broadly describe these relevant techniques from shape analysis including their necessity. To the best of our knowledge, the use of parameterized surfaces in the context of DP has not been explored. The software to accomplish parameterization, registration, and alignment are fully described in (Jermyn et al., 2017), with accompanying implementation at GitHub repository (Laga, 2022). Further, in Figure 10 we display the entire pipeline for our methodology.

### 2.1    PARAMETERIZATION

Each face is realized as a point cloud, a set of $p$ many points in $\mathbb{R}^3$. We describe the data collection process in A.4. A point cloud does not explicitly relay structural information; i.e., there is no natural ordering nor explicit connectivity or relationship between points. Connectivity can be difficult to infer since if two points are close in space, they may not be neighboring points, e.g., two points near the tip of the nose, or any concave feature, can be close in $\mathbb{R}^3$, but based on measuring distance

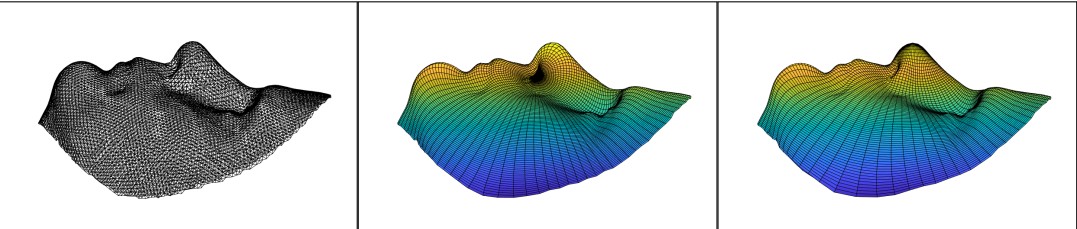

Figure 1: Left: A triangulated mesh face. Middle: A disk-parameterized face, the center of the disk being the left nostril. Right: A disk-parameterized face after a Möbius transformation forcing the center of the disk to be the tip of the nose.

on the surface of the face they may be relatively far apart. So, rather than treat a face as a set of independent points, we can jointly model each face by imposing a parameterization. Specifically, we represent each face as a disk-parameterized surface (Jermyn et al., 2017; Sun et al., 2022; Drira et al., 2010).

We first compute a Delaunay triangulation using the MeshLab software (Cignoni et al., 2008) yielding a triangulated mesh. Given a point cloud $X \in \mathbb{R}^{p \times 3}$ the triangulated mesh, $M = \{V, E, T\}$, is an object which consists of a set of vertices $V$ (the original points), edges $E$, and triangles $T$ composed of said vertices and edges. The triangles meet at edges, do not overlap, and jointly represent a mesh or surface. The left panel of Figure 1 displays a face as a triangulated mesh. We generate a *disk conformal map* of $M$ as in Choi & Lui (2018); a disk conformal map of the triangulated mesh is a mapping from said mesh to a disk which preserves the angles of the triangles or, more generally, the local geometry of the mesh.

Lastly, we generate a disk-parameterized surface as in Jermyn et al. (2017); Laga et al. (2018) via the disk conformal mapping. Figure 1 displays two disk-parameterized surfaces in the middle and right panels; we expand on the differences between these two panels in A.7. Each face, $f$, is thus a map $f : D \to \mathbb{R}^3$ with $D = \{r, \theta | 0 \le r \le 1, 0 \le \theta < 2\pi\}$, the unit disk.

## 2.2 SHAPE ANALYSIS OF SURFACES

Figure 1 displays different representations of a face while retaining its shape. In our setting, the shape of an object is that which is not affected by its scale, location, rotation, or parameterization. The analysis we intend to do should not be dependent on any of these parameters. Next, we describe how to remove this dependence.

Let $\mathcal{F}$ be the space of all parameterized surfaces, $\mathcal{F} \ni f : D \to \mathbb{R}^3$ where $D$ is the unit disk. We assume all surfaces are smooth and genus-0, that is, they are differentiable and have no holes. We first remove scale and location differences by forcing each face to have unit surface area and be centered at the origin. That is, we scale the surface area to be one by setting $f \to f / \int_D |f_r \times f_\theta|_2 dr d\theta$ where $f_r, f_\theta$ are the partial derivatives of $f$ with respect to $r$ and $\theta$, respectively. To center the surface, set $f \to f - \bar{f}$ where $\bar{f}$ is the centroid of the surface. For notational simplicity, let $f$ denote a surface which has unit surface area and the origin as its centroid.

Location and scale are characteristics that are intrinsic to each surface, so we achieve their removal on each face independently. Rotational alignment and parametric registration, however, are relative. Each face must be aligned and registered to a template face.[1] Let $f_{temp}$ be a template face of unit area and origin centroid; in practice one could either choose an arbitrary surface from the dataset, a training set, or some representative surface such as the mean surface.

Let $\mathbb{O} = \{O | \det(O) = 1\}$ be the set of all $3 \times 3$ rotation matrices and let $\Gamma = \{\gamma | \gamma : D \to D\}$ be the set of diffeomorphisms. Each $\gamma$ is a reparameterization of a surface which acts on a surface on the right as $f \circ \gamma$; we fully describe this action in A.3. Rotations, $\mathbb{O}$, act on the left as $O \cdot f$ and do not effect scale nor centroid.

---

[1]In shape analysis it is more typical to do pairwise alignment and registration (Wallace et al., 2014; Cho et al., 2019; Klassen et al., 2004) however our goal is not to do pairwise comparisons so we forego this approach.

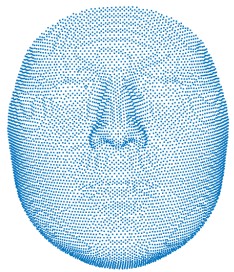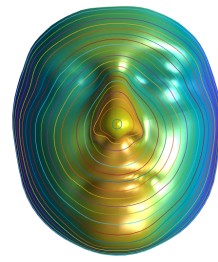

Figure 2: Left: A point cloud representation of a face. Right: A surface representation of the face with an overlay of radial curves.

We optimally align each face to the template, $f_{temp}$, by considering the optimization

$$(\tilde{\gamma}, \tilde{O}) = \mathrm{argmin}_{\gamma \in \Gamma, O \in \mathbb{O}} \|Of \circ \gamma - f_{temp}\|_2.$$

To accomplish this task we appeal to *elastic* shape analysis (ESA) (Jermyn et al., 2017). In short, rather than solve the above minimization, ESA defines a metric which replaces the above $\mathbb{L}^2$ norm and transforms each surface using the *square-root normal field* representation. For each face we compute this minimization such that the registered and aligned face is $\tilde{f} := \tilde{O}f \circ \tilde{\gamma}$. We elaborate on this process in A.3.

## 3   NOVEL FACE RADIAL CURVES REPRESENTATION

We parameterize the entire face using a disk, however, our goal is to work with what we refer to as "face radial curves." This concept is similar to that of *facial curves* (Samir et al., 2006) and *radial curves* (Drira et al., 2010). In the facial recognition literature, for instance, to construct a radial curve representation one picks a focal point, typically the tip of the nose, and overlays curves on the surface of the face with the nose as the center and each point of the curve being equidistant from the tip of the nose. That is, given a focal point $p$ each radial curve is $g = \{x | r = d_F(p, x)\}$ for a given $r \geq 0$ and where $d_F$ is distance measured on the surface of the face.

These previously mentioned methodologies do not take into account that the distribution of features on faces is not uniform for everyone. That is, let $g_{ij}$ represent the $j$th radial curve of the $i$th face, then $g_{ij}$ may be the curve which goes directly on the middle of the eyes for individual $i$ but $g_{i'j}$ may be directly on the forehead of individual $i'$ for the same $j$. Features, such as eyes and the mouth, being disproportionately farther or closer to the focal point or each other hence causes issues. These methodologies do, however consider registration within each $j$ but not across all $j$s simultaneously. We propose a new method to have an entire global alignment and registration across all sets of curves and faces.

In the previous section, we set up a way to compute faces $\{\tilde{f}^{(i)}\}_{i=1,\cdots n}$ that are registered and aligned to a template face $f_{temp}$. Each face is a mapping $\tilde{f}^{(i)} : D \to \mathbb{R}^3$ and we note a disk is an infinite collection of concentric circles, and thus each $\tilde{f}^{(i)}$ is a collection of curves $\tilde{f}_r^{(i)} : D_r \to \mathbb{R}^3$, where $D_r \subset D$ is the circle of radius $r$. By construction, the alignment and registration to a template implies these curves $\tilde{f}_r^{(i)}$ capture the same features across all faces; these curves are what we refer to *face radial curves*.

We apply our proposed method to data described in Sero et al. (2019); more details are available in A.4. The left panel of Figure 2 displays a face as a point cloud and the right panel is the same face but with the disk parameterization and some of the face radial curves overlain. While the disk parameterization is an infinite collection of these curves, we pick some number of these curves to represent the face. The number of curves is a tuning parameter where more curves implies, to an extent, more definition in the face. We discuss this further in 5.

## 4 GAUSSIAN DIFFERENTIAL PRIVACY FOR FUNCTIONAL DATA

Since the conception of DP (Dwork et al., 2006), noise calibration has been considered from different perspectives leading to many variants. For instance, "zero concentrated differential privacy" (Bun & Steinke, 2016) and "Rényi differential privacy" (Mironov, 2017) are notions of DP from the perspective of a divergence of the distribution of the mechanism. In the present paper we utilize "Gaussian DP" (GDP, Dong et al. (2019)), which considers DP from the perspective of a particular set of hypothesis tests. As one of our key contributions in this paper, we extend GDP to functional data analysis (FDA) in Section 4.2. We begin with a brief overview of DP for FDA.

### 4.1 BACKGROUND ON DIFFERENTIAL PRIVACY FOR FUNCTIONAL DATA

In this section and in A.5, we describe approximate DP, $(\epsilon, \delta)$-DP, in the context of FDA. This is a needed background for our proposed extension of GDP into the space of functional data. For a more thorough exposition on DP, see Blum et al. (2005); Dwork & Roth (2014); Dwork et al. (2006).

Let $D$ denote a dataset of a size $n$, $D = \{x_1, x_2, \ldots, x_n\}$, and $\mathcal{D}$ be the space of all such datasets. An *adjacent* dataset, $D'$, is a dataset which differs from $D$ in exactly one element which, without loss of generality, we can choose as the last element, $D' = \{x_1, x_2, \ldots, x_n'\}$. We write $D \sim D'$ to denote adjacency. Here $h(D)$ denotes the statistical summary we aim to release, and a private random version of the summary as $\tilde{h}(D)$, which we will refer to as a privacy mechanism.

We consider releasing an estimate of a private mean function. To sanitize functions we rely on tools and foundations of functional data analysis in the domain of privacy as in Hall et al. (2013); Alda & Rubinstein (2017); Mirshani et al. (2017), with the latter considering spaces more extensive than functions. The infinite dimensional nature of function valued summaries presents a critical challenge in establishing formal privacy. In particular, traditional probability densities become much more complicated as there is no default or baseline measure in infinite dimensions (unlike Lebesgue measure in $\mathbb{R}^d$). To overcome this challenge, there are currently two approaches. The first, taken by Hall et al. (2013) is to work in finite dimensions and then take careful limits. The second approach, introduced in Mirshani et al. (2017) and which we follow here, is to utilize carefully constructed infinite dimensional densities so that the probability inequalities can be worked out directly (and avoid having to take limits).

Let $\mathbb{H}$ denote a real separable Hilbert space in which we aim to release a summary statistic $h(D) \in \mathbb{H}$, e.g., $\mathbb{L}^2([0,1])$, $\mathbb{R}^n$, or a reproducing kernel Hilbert space. We consider a Gaussian process in $\mathbb{H}$ to add noise to $h(D)$. Let $X$ be a Gaussian process in $\mathbb{H}$ parameterized by its mean $\eta \in \mathbb{H}$ and covariance operator $C : \mathbb{H}^* \to \mathbb{H}$ where $\mathbb{H}^*$ is the dual space of $\mathbb{H}$. A stochastic process is said to be Gaussian if any linear functional $a \in \mathbb{H}$ of $X$ is Gaussian in $\mathbb{R}$. We note that technically $a \in \mathbb{H}^*$, however since $\mathbb{H} \cong \mathbb{H}^*$, we avoid this distinction unless necessary. Further, we have that $\mathbb{E}[a(X)] = a(\eta)$ and $C(a, b) = \text{Cov}(a(X), b(X))$ with $a, b \in \mathbb{H}$. We write that $X \sim \mathcal{N}(\eta, C)$ and $Z \sim \mathcal{N}(0, C)$. In this setting we can achieve approximate DP via Theorem A.2. A critical requirement is that our summary be *compatible* with the noise $Z$ which, roughly stated, means that while the noise lives in $\mathbb{H}$, we require our summary to exists in a smaller space $\mathcal{H} \subset \mathbb{H}$. In our case $\mathcal{H}$ is a reproducing kernel Hilbert space (RKHS) so the eigenfunctions arise from $C$ and thus are determined by our choice of kernel $k(s, t)$.

Now that we have well defined noise, we consider releasing a specific summary statistic: a private mean function with sample mean $h(D) = \bar{X} = \frac{1}{n} \sum_i x_i$. To have some control on the smoothness of this estimate, we enforce smoothness by relying on an optimization problem with a penalty. That is, we let $h(D) = \text{argmin}_{m \in \mathcal{H}} \sum_i \|x_i - m\|_{\mathbb{H}}^2 + \phi \|m\|_{\mathcal{H}}^2$ with $\phi$ being a smoothness penalty parameter. This approach ensures the required noise compatibility. For our summary statistic, the sensitivity, $\sup_{D \sim D'} \|h(D) - h(D')\|^2$, can be shown to be bounded as $\Delta^2 \le 4\tau^2/(n^2\phi)$ (Mirshani et al., 2017), where $\tau$ is a finite bound on the norm in the $\mathbb{H}$ space of the elements of all $D \in \mathcal{D}$.

### 4.2 EXTENSION OF GDP TO FUNCTIONAL DATA

One especially useful interpretation of approximate DP comes from Wasserman & Zhou (2010), which relates DP to hypothesis testing. In particular, for a sanitized output, $\tilde{h}(D)$, one can consider statistical tests for determining if the true underlying data source is $D$ or some adjacent data set $D'$.

In these simple cases, the optimal test is well known (Neyman-Pearson Lemma), so one can talk about the optimal type 2 error (1-power) of a statistical test for given type 1 error rate. It turns out that DP gives a bound for the type 2 error relative to the type 1 error, which Dong et al. (2019) took a step further to define Gaussian DP. In particular, Dong et al. (2019) show a mechanism is $\mu$-GDP if its entire type 1/type 2 error tradeoff curve is bounded by that of a curve coming from comparing a $N(0,1)$ to $N(\mu,0)$, meaning it is harder to distinguish 0 and $\mu$ from $N(0,1)$ than it is $h(D)$ from $h(D')$ from $\tilde{h}(D)$. We provide the formal definition below.

**Definition 4.1.** A mechanism $\tilde{h}(D)$ is said to satisfy $\mu-$Gaussian differential privacy ($\mu-$GDP) (Dong et al., 2019) if for all adjacent datasets $D \sim D'$,

$$T(\tilde{h}(x;D), \tilde{h}(x;D')) \geq G_\mu,$$

where $T$ is a trade-off function and $G_\mu := T(N(0,1), N(\mu,1))$. Here a trade-off function $T : [0,1] \to [0,1]$ for two probability distributions $U_1$ and $U_2$ is $T(U_1, U_2)(\alpha) = \inf_\zeta \{\beta_\zeta : \alpha_\zeta \leq \alpha\}$ with $\zeta$ being a rejection rule, $\alpha$ being the type 1 error, and $\beta_\zeta$ the type 2 error for $\zeta$.

In our main Theorem 4.3, we prove that the DP framework for functions as defined in Theorem A.2 is $\mu-$GDP; we utilize the following corollary to achieve this.

**Corollary 4.2** (Dong et al. (2019)). *A mechanism is $\mu-$GDP if and only if it is $(\epsilon, \delta(\epsilon))-$DP for all $\epsilon \geq 0$, where $\delta(\epsilon) \geq \Phi\left(-\frac{\epsilon}{\mu} + \frac{\mu}{2}\right) - e^\epsilon \Phi\left(-\frac{\epsilon}{\mu} - \frac{\mu}{2}\right).$*

Corollary 4.2 also appears in Balle & Wang (2018) in the context of the calibrating the Gaussian mechanism, but in the presented form the corollary is more readily applicable for our setting.

**Theorem 4.3.** *The mechanism $\tilde{h}(D) = h(D) + \sigma Z$ as defined in Theorem A.2 is $\mu$-GDP with $\sigma \geq \Delta/\mu$. Here $\Delta$ is the same global sensitivity as before.*

*Proof.* We need to show that the distribution induced by our Gaussian process in $\mathbb{H}$ of $\tilde{h}(D)$ is $\mu-$GDP by bounding the $\delta(\epsilon)$ in Theorem A.2 as in Corollary 4.2. The key here is the use of $\mathbb{H}$, as our function space is infinite-dimensional and hence it is not possible to define a measure analogous to that of the Lebesgue measure.

Mirshani et al. (2017) provide the framework to define a useful density of $\tilde{h}(D)$ over $\mathbb{H}$, however they do not consider bounding the tail probabilities nor the privacy loss random variable as we do next. We first define the privacy loss random variable as $PL = \log\left[\frac{dP(D)}{dQ} / \frac{dP(D')}{dQ}\right]$, where $Q$ is the probability measure induced by our noise $Z$ and $P(D)$ the family of measures of our mechanism $\tilde{h}(D)$. By construction we have that $h(D)$ is compatible with $Z$ and hence the above is well defined:

$$\frac{dP(D)}{dQ}(y) = \exp\left\{-\frac{1}{2\sigma}\left[\|h(D)\|_\mathcal{H}^2 - 2T_{h(D)}(y)\right]\right\}.$$

Here $T_{h(D)}$ is a linear operator and thus $T_{h(D)}(\tilde{h}(D))$ is normally distributed.

From Mirshani et al. (2017), we have that $T_{h(D)}(\tilde{h}(D)) - T_{h(D')}(\tilde{h}(D)) \sim N(0, \|h(D) - h(D')\|_\mathcal{H}^2)$. Using this, we can show that

$$\delta := P(PL \geq \epsilon) - e^\epsilon P(PL \leq -\epsilon) \tag{1}$$

$$= \Phi\left(-\frac{\epsilon\sigma}{\Delta} + \frac{\Delta}{2\sigma}\right) - e^\epsilon \Phi\left(-\frac{\sigma\epsilon}{\Delta} - \frac{\Delta}{2\sigma}\right) \tag{2}$$

$$= \Phi\left(-\frac{\epsilon}{\mu} + \frac{\mu}{2}\right) - e^\epsilon \Phi\left(-\frac{\epsilon}{\mu} - \frac{\mu}{2}\right). \tag{3}$$

The last equality is a reparametrization setting $\mu = \Delta/\sigma$ and the rest of the details are in the A.8. $\square$

For facial radial curves and other disc-like surfaces, by construction, we need to sanitize many mean curves, so we require composition of a multitude of mechanisms with their respective budgets $\mu_i$'s. Composition in the GDP framework is "tight." Given two mechanisms $\tilde{h}_1, \tilde{h}_2$ with privacy parameters $\mu_1$ and $\mu_2$, their composition $(\tilde{h}_1, \tilde{h}_2)$ is $\sqrt{\mu_1^2 + \mu_2^2}$-GDP (Dong et al., 2019, Corallary 3.3).

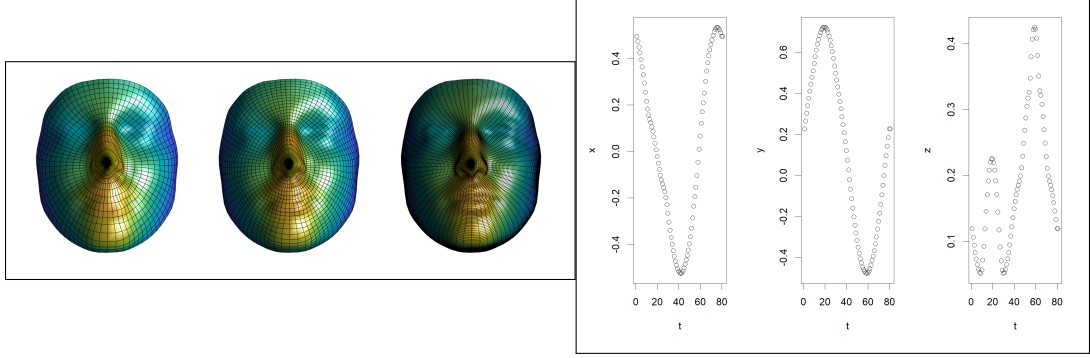

Figure 3: Left: The same face represented using 16, 27, and 80 face radial curves, respectively. Right: The $x$, $y$, and $z$ coordinate curves, respectively, for a particular face radial curve.

## 5  FACE RADIAL CURVE EXAMPLE

Here, we apply our proposed methods from sections 3 and 4.2 on data described in Sero et al. (2019) and A.4. Figure 3 displays from left to right the same face with $J = 16$, 27, and 80 face radial curves. Larger values of $J$ add definition to the face, but $J$ need not be too large to encapsulate the facial structures. Based on that, we represent the $n = 1000$ faces with $J = 23$ face radial curves; one could pick $J$ in a data driven way as well. We treat the $J$ sets of curves across the faces independently.

We note an important subtle strength in our construction. Each face radial curve has three coordinates, $x, y, z$, and by our construction $x, y$ are effectively a circle and $z$ encode facial features. The right panel of Figure 3 displays the coordinate curves of an example face radial curve. The first two coordinates are quite simple and are roughly one period of a sine curve. We leverage this simplicity in these two coordinate curves to conserve privacy budget by enforcing a larger smoothing parameter as compared to the last coordinate curve and also treat the coordinate curves separately at each radial curve.

Let $\{f_i\}$ be the set of radial curves for a specific coordinate at the $j$th position and $i$ being the index of for particular face. For simplicity one can imagine $j = 1$ being the curve nearest the tip of the nose and $j = 23$ as the curve nearest the border of the face. Each curve $f_i$ is a closed parameterized curve, $f : \mathbb{S}^1 \to \mathbb{R}$ where $\mathbb{S}^1$ is the unit circle, as in the right panels of Figure 3. In general, we drop the $j$ as we treat each set independent of each other. We parameterize the curve with unit circle but for simplicity we say $f : [0, 1] \to \mathbb{R}$ with $f(0) = f(1)$.

By design we have closed curves, so to retain this structure we use a periodic kernel that takes the form $k(s, t) = \exp\left(- \left[d_{\mathbb{S}^1}(\omega(t), \omega(s))/\rho\right]^\alpha\right)$ (Gneiting, 2013) with $t, s \in [0, 1]$. The distance of two points $a, b$ on $\mathbb{S}^1$ is $\arccos\langle a, b\rangle$, however, the previous $t, s$ are parameters on the unit interval, so we first "wrap" the interval around the circle to compute this distance hence the need for $\omega(\cdot)$, a wrapping function. This kernel is a powered exponential on spheres with kernel range parameter $\rho$ and smoothness parameter $\alpha \in (0, 1]$. For our experiments we set $\alpha = 1$ as we can control smoothness instead via $\phi$ in the mean estimation as in 4.1.

We compute kernel matrix $K$ of the the unit interval $[0, 1]$ with $[K]_{ij} = k(\omega(i/m), \omega(j/m))$ where $m = 80$ is an integer defining the fineness of the uniform grid on the unit interval and $i, j = 0, 1, \ldots m$. Let the eigenvalues, of $K$, and their associated eigenfunctions be denoted as $(\lambda_i, b_i)$. It can be shown that non-private mean in the RKHS space is $h(D) = \frac{1}{n} \sum_i \sum_j \frac{\lambda_j}{\lambda_j + \phi} \langle f_i, b_j\rangle b_j$. Since we are sanitizing the coordinates independently, we have a smoothness parameter for each coordinate, $\phi_x$, $\phi_y$, and $\phi_z$. The left panel of Figure 4 displays the mean face constructed from mean face radial curves with $\phi_x = \phi_y = 0.01$ and $\phi_z = 0.005$.

### 5.1  EXPERIMENTAL RESULTS

To highlight the difficulty of working with such complex data we present some preliminary results. We have 1000 faces each with 7150 points; we fully describe the data in A.4. The points are

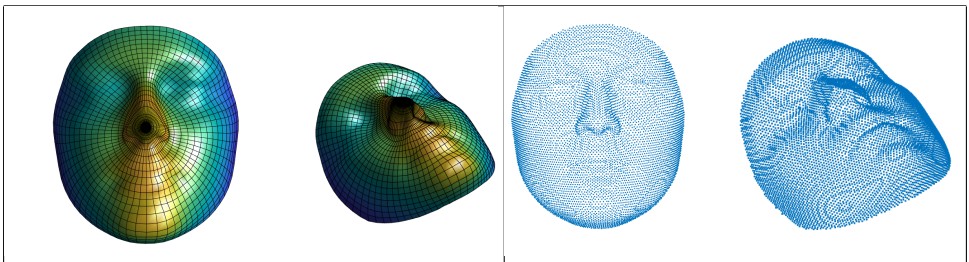

Figure 4: Left: Two angles of the average face constructed of average face radial curves with $\phi_x = \phi_y = 0.01$ and $\phi_z = 0.005$. Right: Two angles of the point-wise mean.

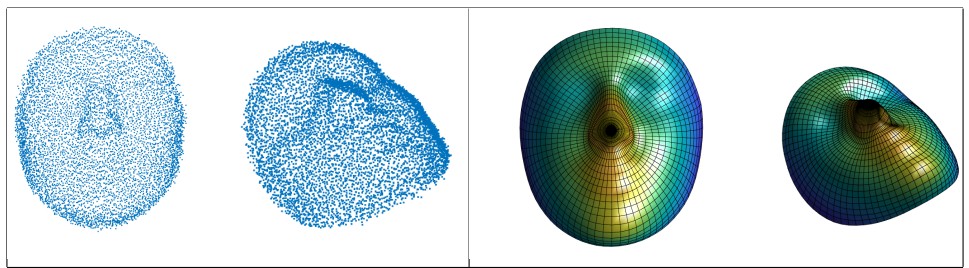

Figure 5: Left: Two angles of a private face sanitized with point wise Gaussian noise with total $\mu_T = 2$. Right: Two angles of a differentially private face constructed from private mean face radial curves with $\mu_T = 2.9961$ $\phi_x = \phi_y = 0.01$ and $\phi_z = 0.005$.

"registered", via the data collection method, across all the faces. Without this registration, the following would not be immediately feasible. Let $D = \{X_i\}$ and $X_i \in \mathbb{R}^{7150 \times 3}$. We first compute the point-wise mean of the faces, $\bar{X} = 1/n \sum_i X_i$ and display this in the right panel of Figure 4.

Suppose we have a total budget of $\mu_T$ and we wish to sanitize, independently, the 7150 points each of which has 3 coordinates. To split the budget evenly, each point's coordinate is allocated $\mu_p := \sqrt{\mu_T^2/(7150 \cdot 3)}$ of the budget. For each point and each coordinate we compute sensitivity as $\Delta_{k,l} = \max_{i,j} |X_i[k,l] - X_j[k,l]|$. That is, among all faces this measures the variability at each point. We note that this sensitivity calculation does violate privacy, as it is data driven, and that an entirely private form of this calculation is strictly larger. That is, our sensitivity calculation is smaller, so a private method would lead to noisier estimates. We add noise to each coordinate of each point as $\bar{X}[k,l] + \Delta_{k,l}/\mu_p \cdot z$ where $z \sim \mathcal{N}(0,1)$. Figure 5 displays two angles of a point-wise private face in the left panel with total privacy budget $\mu_T = 3$ with more results in A.9.

Next we apply our approach using face radial curves. We have $J$ many sets of curves, $\{f_i\}_j$ each of which has three coordinate curves, $f_i = (f_{ix}, f_{iy}, f_{iz})$. We sanitize the regularized mean, independently, for each coordinate and each radial curve, using our mechanism which satisfies GDP with sensitivity is $\Delta^2 \leq 4\tau^2/(n^2\phi)$. Earlier we saw that the $x$ and $y$ coordinate curves, at every $J$, are effectively one period of a sinusoidal curve, we take advantage of this construction to conserve budget. We thus spend less budget sanitizing the $x$ and $y$ coordinate curves and spend more on $z$ as they contain more feature information.

We need to determine the $\tau$, an upper bound on the norm of the curves, in our $\Delta$. We determine this $\tau$, in a similar way as the point-wise approach of computing sensitivity, through the data at each set of radial curves and each coordinate. That is, for each $j$ and at each coordinate $w$, $\tau_w = \max_i \|f_i(w)\|_{\mathbb{H}}$. Again, we have a sensitivity that is partially data driven. However we note since we use shape analysis to do our processing, all the faces are scaled to have unit surface area, and hence all the curves are scaled. Thus, individually none of these norms hold any meaningful size information. Further, our analysis is entirely *scale invariant*, meaning that if a face is much larger or smaller than average, that information is completely removed when we scale. This is a strength of shape analysis as the emphasis is the *shape* of the face and not the nuisance parameters.

Table 1: Mean Squared Error between private estimates and the point-wise mean. The last column is the point-wise MSE between our method and the RKHS mean. All values are at E-04.

| | | $\mu_T$ | | |
|---|---|---|---|---|
| | 2 | 3 | 2.9961 (Ours) | Ours* |
| MSE | 14 | 7.5807 | 5.2989 | 3.6365 |

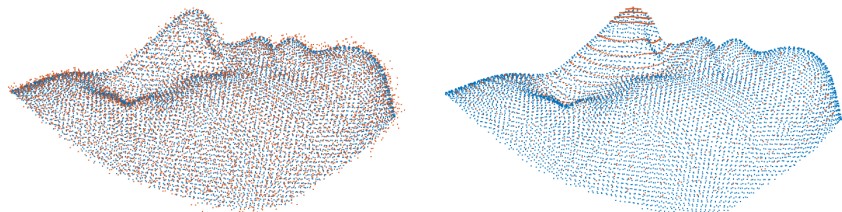

Figure 6: Blue points are the point-wise mean. Left: The red points are the point-wise private mean with $\mu_T = 3$. Right: The red points are our private mean with $\mu_T = 2.9961$.

We choose a budget for each of the coordinates $\mu_x, \mu_y, \mu_z$ which are the same at all levels $j$ of curves; i.e., at each $j$ for the $w$ coordinate curve $\tilde{h}(\{f_{iw}\}) = h(\{f_{iw}\}) + \frac{\Delta_w}{\mu_w} Z$ with $\Delta_w^2 \leq 4\tau_w^2/(n^2\phi)$ and $Z$ is a standard normal Gaussian process. The right panel of Figure 5 displays an example private mean face with $\mu_T = 2.9661$ where $\mu_T = \sqrt{J(\mu_x^2 + \mu_y^2 + \mu_z^2)}$ with $J = 23$, $\mu_x = \mu_y = 0.2$ and $\mu_z = 0.55$. For each coordinate we have separate smoothing parameters we display results with $\phi_x = \phi_y = 0.01$ and $\phi_z = 0.005$. Comparing our private mean to left panel of Figure 4 we see there is some clear smoothing in the lips area and some subtle smoothing in the eyes as well.

Lastly, we quantify the amount of injected noise by computing a mean squared error. We let the point-wise mean be our baseline non-private estimate, the right panel of Figure 4. For the point-wise private estimate $\tilde{X}$ the $MSE = \frac{1}{N} \sum_i |\bar{X}_i - \tilde{X}_i|^2$ where $N = 7150$ is the number points. Table 1 displays these errors in the first two columns. Our sanitized mean, however, is a set of functions so we first discretize it into 1863 points (23 curves at 81 points) and denote this as $\tilde{V}$. We let $MSE = \frac{1}{M} \sum_j \min_i |\bar{X}_i - \tilde{V}_j|^2$ with $M = 1863$. Since the two point clouds have different number of points, this MSE finds the nearest point in the non-private face to that of the private face $\tilde{V}$. We further scale $\tilde{V} \to a\tilde{V}$ to align it to $\bar{X}$ as $\bar{X}$ is not processed data. Table 1 displays the error in the third column of 5.2989, less than both of the point-wise private faces. This MSE does incur an inflation, though, as the point-wise mean is not a surface and thus lacks a registered point at the location of our private mean. We can clearly see this in Figure 6, in both panels the blue points are the point-wise mean, the left panel has the point-wise private mean in red, and the right panel has our private (discretized) mean in red. We see that our method has points that seem to lie on the "surface" of the face but the non private mean may not have a point there. We also clearly see that the point-wise private mean adds noise in the ambient space and thus creates a rough, fuzzy estimate not entirely resembling a smooth face; i.e., the facial structures are distorted. Also, the MSE of our facial radial curve based GDP method is less than the MSE of the point-wise sanitized estimate which has a slightly larger privacy budget. To even further emphasize this point, the last column of Table 1 is the MSE between our private mean and the RKHS mean, we see it is less than half in comparison to the point-wise sanitization.

## 6  CONCLUSIONS

We have developed a framework for releasing a $\mu$-GDP human face with our novel face radial curve representation. We extended $(\epsilon, \delta)$-DP FDA (Mirshani et al., 2017) techniques into the $\mu$-GDP framework (Dong et al., 2019) to take advantage of its tight composition of budgets. We utilized the new $\mu$-GDP FDA framework specifically for faces but note this is applicable for other FDA applications. Further, we utilize the shape analysis techniques of Samir et al. (2006); Drira et al.

(2010); Jermyn et al. (2017) to create a novel curve representation of a face. We focus on human faces, however, one can use our framework to sanitize other surfaces which are diffeomorphic to a disk. Our representation changes the problem of sanitization from needing thousands of point-wise estimations to a few dozen functional estimates. We discuss the limitations of our methods in A.1.

A persons face contains information of ones identity and, as noted earlier, its image contains genomic information (Venkatesaramani et al., 2021), thus, the need to protect this information while sharing these data (e.g., anthropological studies) is crucial. We demonstrate via empirical and quantitative results that our methodology adds less noise and preserves the structure of the face. We chose the number of face radial curves needed for our representation, but one could develop a private cross-validation method to do so.

In A.4 we describe how the data is collected by Sero et al. (2019), so here we mention some limitations in our experimental results as well as in A.1. In a sense, our data is very clean, and hence our methodology is only tested in this circumstance. All faces in the dataset have a neutral expression, so to handle other expressions would require additional processing or extending the methodology. Expression variation is a complex source of variability which is critical in areas such as face recognition (Smeets et al., 2012). It is not entirely clear how to handle such variability for estimation of an average. The parametrization requires a genus-0 surface, i.e. no holes, so if a face has missing data this would need to be rectified. For instance, preprocessing by patching "holes" or missing data has been considered by Passalis et al. (2011) which leverage the symmetry of the face for interpolation. There is a vast amount of survey papers in the face cognition literature such as Zhao et al. (2003); Jafri & Arabnia (2009); Li et al. (2020), to name a few, which encounters similar issues all to say these problems are not trivial and are potential future research opportunities.

## ACKNOWLEDGMENTS

The authors would also like to thank the anonymous reviewers which provided constructive feedback. CS would also like to thank Gary Choi for guidance on conformal maps. This work was in part supported by NSF Award No. SES-1853209 to the Pennsylvania State University, and by Huck Institutes of the Life Sciences at the Pennsylvania State University through the Dorothy Foehr Huck and J. Lloyd Huck Chair in Data Privacy and Confidentiality. Content is the responsibility of the authors and does not represent the views of the Huck Institutes.

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

# A Appendix

## A.1 Limitations

We do not collect any data, however, in A.4 we describe the data and how it was collected. In many regards our data is clean, so we did not test our methods on noisy data. In a sense, we can consider our data as processed data, so noisy data would require some processing which we do not consider in this work. It is possible that if the data is more noisy then it would require further smoothing which our method does allow. We further have an assumption of a genus-0 surface (a face with no holes) which may be an issue if there are missing data points. This type of data would require additional pre-processing. We also only test our method on one dataset; one could encounter the issue of variability of noise in other datasets but this can be overcome with the smoothness parameters in our method.

A limitation which we have already mentioned in 5.1 is the method for computing sensitivity, $\Delta$, in both our method and the point-wise method which we compare ourselves to. This, however, is an inherent issue in differential privacy rather than our method. That is, for both our method and the point-wise private estimate, we use the data to estimate the sensitivity. Ideally one could use a second dataset or a training set to estimate the many sensitivity values needed in a private way.

In terms of limitations of privacy and fairness, these are the exact problems we address. As mentioned in 1, one may want to release private average faces for demographics. Our methods are intended for facilitate this goal although we do not consider variability of sample sizes by demographics.

## A.2 Experiments Compute Resources

We ran experiments almost exclusively on a desktop computer with an Intel i7 processor and 32GB of RAM on Windows 11. Creating a triangulated mesh for a face takes about 5 seconds on the

desktop, so this was done locally for each face. Parameterizing each face costs about 20 seconds on the desktop, this was also done locally. Reparameterizing, however, each face to a template costs upwards of 3 minutes on the desktop, so since we have 1000 faces we did this on a supercomputer.

Computing the mean and privatization with our method costs about 3 total minutes on the desktop computer. The mean computation and privatization of the point-wise method only requires about 1 minute of computational time on the desktop.

## A.3 REPARAMETERIZATION

To optimally register two surfaces we use the methods as described in Jermyn et al. (2017) which we summarize next. Suppose we have two surfaces $f_1, f_2 \in \mathcal{F}$ which we wish to optimally register over the parameterization group $\Gamma$. Here $\mathcal{F}$ is as in 2.2 $\mathcal{F} \ni f : D \to \mathbb{R}^3$ where $D$ is the unit disk. The action of $\Gamma$ on a surface is right composition, $f \circ \gamma$ for $\gamma \in \Gamma$, and reparameterizes the surfaces but does not change its image which in our case is the face. The authors in Jermyn et al. (2017) leverage a transformation of $f$ referred to as the square-root normal field (SRNF) defined as $q = \frac{n}{|n|^{1/2}}$ where $n$ is the normal vector $n = \frac{\partial f}{\partial u} \times \frac{\partial f}{\partial v}$. The corresponding action of $\Gamma$ on $q$ is then $(q, \gamma) := \sqrt{J(\gamma)} q \circ \gamma$. This transformation is necessary for an isometric action of $\Gamma$ but the details of this are lengthy and not necessary for our application.

The authors define an energy function $E_{reg} : \Gamma \to \mathbb{R}_{\geq 0}$ to implement an iterative gradient descent method for the registration. The energy is defined as $E_{reg}(\gamma) = \|q_1 - (\tilde{q}_2, \gamma)\|^2$ where $\tilde{q}_2$ is the "current" stage of the surface being reparameterized. Here $\gamma$ is the incremental reparameterization with $\tilde{q}_2 = (q_2, \gamma)$. Here $q_1$ would be our template surface from 2.2 and we register all surfaces to this template.

Since the reparameterization is done in an *iterative* manner it results in incrementally improved registration. This method is iterative as the gradient is taken about the identity of $\Gamma$, $\gamma_{identity}$ and thus lives in the tangent space of this element. Suppose we have an orthonormal basis $\mathcal{B} = \{b\}$ for the tangent space of the $\Gamma$ at the identity, $T_{id}(\Gamma)$ where each $b$ is a unit "vector" in the tangent space $T_{id}(\Gamma)$. The full gradient is given by $\sum_{b_i \in \mathcal{B}} \langle q_1 - \tilde{q}_2, d(b_i, \gamma) \rangle_2 b_i$ and we use the implementation mentioned in 2.2 and refer the interested reader to the cited materials.

This reparameterization is costly due to the dimension, iterative construction, and reliance on a basis. To alleviate some expense, we set the center of the surface as the tip of the nose as noted in A.7 with the triangulated mesh before we impose the disk parameterization. The reparameterization iterative method thus has to search over a smaller space as the center of the disk is pre-registered.

## A.4 DATA DETAILS

For a full description of how the data is collected we refer to Sero et al. (2019) but we summarize and emphasize the relevant points. All participants have the same neutral face expressions, so the data is not heterogeneous in terms of facial expression. Each face is captured with 7150 points which the authors refer to as quasi-landmarks. Further, these landmarks are registered across all individuals which Sero et al. (2019) refer to as "homologous." This registration is a point-wise correspondence across faces while the registration in A.3 is an entire disk correspondence. Similar to our approach the faces are aligned using Procrustes Analysis, which finds the optimal rotation but again considering the faces as a set of points not a disk-like surface.

## A.5 SUPPLEMENTAL NOTES ON DP

We first present the definition for approximate differential privacy.

**Definition A.1** ((Dwork et al., 2006))**.** Let $D \sim D'$ and $\tilde{h}(D)$ be a random privacy mechanism, the mechanism is said to achieve approximate differential privacy, $(\epsilon, \delta)$-DP, for some $\epsilon > 0$ and $0 < \delta < 1$, if it satisfies the probabilistic inequality

$$P(\tilde{h}(D) \in A) \leq e^\epsilon P(\tilde{h}(D') \in A) + \delta,$$

for any measurable set $A$.

Both $\epsilon$ and $\delta$ are pre-specified parameters referred to as the privacy budget. When $\delta = 0$, this is referred to as *pure* differential privacy. Differential privacy is an attribute of the random mechanism and roughly states that the distributions over adjacent datasets are not too different. To achieve approximate DP for functional data Mirshani et al. (2017) establish the following mechanism.

**Theorem A.2** (Mirshani et al. (2017)). *Let $h(D)$ be a functional summary of a dataset $D$ that is compatible with standard Gaussian process noise $Z$ and $\epsilon \leq 1$. We have that $\tilde{h}(D) := h(D) + \sigma Z$ achieves $(\epsilon, \delta)-DP$ over $\mathbb{H}$ where $\sigma \geq \frac{2 \log(2/\delta)}{\epsilon^2} \Delta^2$. Here $\Delta$ is the global sensitivity of the summary $h(D)$ and $\Delta^2 = \sup_{D \sim D'} \|h(D) - h(D')\|_{\mathcal{H}}^2$ where the norm is over the space of the noise $Z$.*

Assuming compatibility, let $Q$ denote the probability measure induced by $Z$ over $\mathbb{H}$ and $\{P(D) : D \in \mathcal{D}\}$ denote the family of measures over $\mathbb{H}$ induced by $\tilde{h}(D)$ as in Theorem A.2. The density of $\tilde{h}(D)$ over $\mathbb{H}$ with respect to $Q$, which exists if compatability holds, takes the form

$$\frac{dP(D)}{dQ}(y) = \exp\left\{-\frac{1}{2\sigma}\left[\|h(D)\|_{\mathcal{H}}^2 - 2T_{h(D)}(y)\right]\right\},$$

$Q$ almost everywhere with $T_{h(D)}(y) = \langle h(D), y\rangle_{\mathcal{H}}$. The inner product on $\mathcal{H}$ can be defined in terms of eigenvalues ($\lambda_i$) and eigenfunctions ($b_i$), as $\langle x, y\rangle_{\mathcal{H}} = \sum_i \lambda_i^{-1} \langle x, b_i\rangle_{\mathbb{H}} \langle y, b_i\rangle_{\mathbb{H}}$ but can generally be expressed using any basis.

## A.6 PRIVATE POINT-WISE FACE

Here we have more details on the benchmark method. We have $n = 1000$ faces each with 7150 points. Let $D = \{X_i\}$ and $X_i \in \mathbb{R}^{7150 \times 3}$ be the set of faces; further let $X_i[k, l]$ be the $l$th coordinate, $(x, y, z)$, of the $k$th point of the $i$th face, $l \in \{1, 2, 3\}$, $k \in \{1, \ldots, 7150\}$, and $i \in \{1, \ldots, 1000\}$. The points are "registered" across all the faces i.e. $X_i[k, \cdot]$ and $X_j[k, \cdot]$ represent the same facial feature e.g. the tip of the nose.

We compute the point-wise mean of $D$, the faces, as $\bar{X} = 1/n \sum_i X_i$ and display this in the right panel of Figure 4. We have a total budget of $\mu_T$ for the entire face which we need to split across all points, $\bar{X}[k, \cdot]$. We sanitize, independently, the 7150 points, $\bar{X}[k, \cdot]$, each of which has 3 coordinates $\bar{X}[k, l]$. To split the budget evenly, each point's coordinate is allocated $\mu_p := \sqrt{\mu_T^2/(7150 \cdot 3)}$ of the budget. That is, we require $(7150 \cdot 3)$ many mechanisms and their budget composition is $\sqrt{\sum \mu_p^2}$, since we are using $\mu$-GDP. For each coordinate of each point of the average, $\bar{X}[k, l]$, we compute sensitivity as $\Delta_{k,l} = \max_{i,j} |X_i[k, l] - X_j[k, l]|$. We add noise to each coordinate of each point as $\bar{X}[k, l] + \Delta_{k,l}/\mu_p \cdot z$ where $z \sim \mathcal{N}(0, 1)$.

## A.7 MÖBIUS TRANSFORMATION

The reparameterization defined in 2.1 is computationally expensive. To make the repameterization less expensive, we leverage the Möbius transformation of conformal maps. We give a high level idea of this transformation with more details available in Choi & Lui (2015; 2018); Choi et al. (2020) and implementation available at GitHub repository Choi (2020).

The left most panel of Figure 7 displays a triangulated mesh of a face. The next step in our pre-processing is to generate a disk conformal map. The middle column of Figure 7 displays two different disk conformal maps of the same triangulated mesh in the left panel. It is, admittedly, difficult to discern features, but in each panel of the middle column one can see approximately four dense areas of which the middle corresponds to the nose. Recall that this disk conformal map embeds the triangulated mesh onto the disk while attempting to locally preserve all angles.

The panels in the right column of Figure 7 are the disk parameterized surface corresponding to the adjacent middle column disk conformal map. The center of the disk conformal disk, and hence the disk parameterized surface, is not known a priori. Either of these disk parameterized surfaces will suffice in our construction as we can optimally register it to a template as explained in 2.2. However, since reparameterization is costly, we can save cost by prespecifying the center of the disk of both the template and each face. A natural choice is to designate the center of the disk with feature such as the tip of the nose.

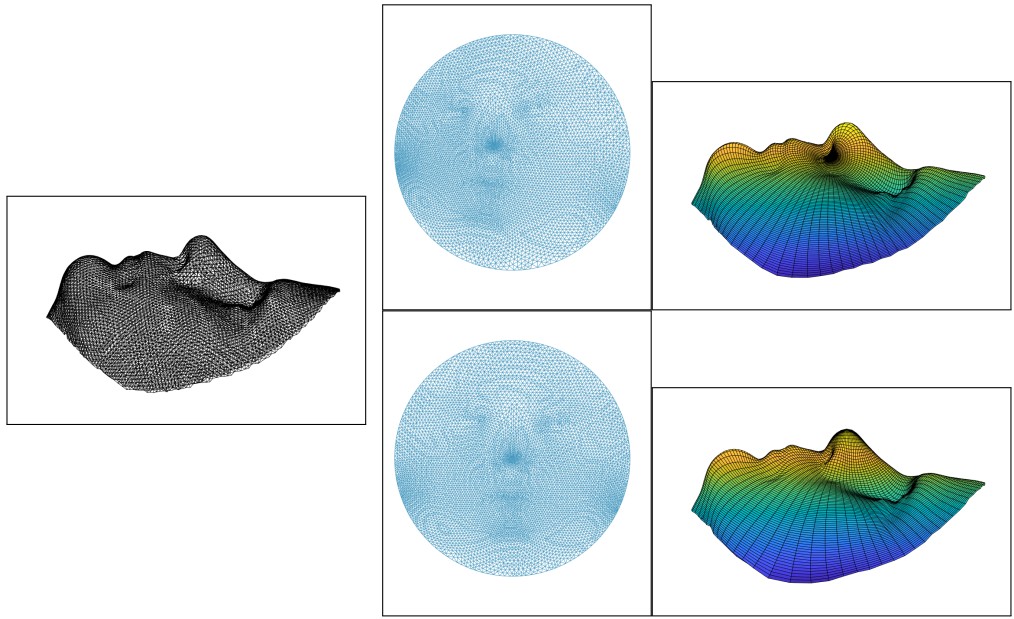

Figure 7: Left column: A triangulated mesh of a face. Middle column: Two disk conformal maps of the triangulated mesh on the left. Right column: The disk parameterized surface resulting from the adjacent middle column conformal map.

The bottom panel of the middle row is constructed using the Möbius transformation on the disk conformal map of the top panel of the middle row. We note that the Möbius transformation is for the disk conformal map and not the disk paramterized surface. This Möbius transformation costs effectively no time, but reparameterization of the bottom right disk parameterized surface is much faster than reparameterizing the top right to the template. This is because the methodology in A.3 looks at the gradient about the identity, so shifting the center of the disk requires a lot of energy. Having prespecified the nose as the center of the disk, though, the search space for reparamterization is much smaller. We lastly also note that all from the left and right columns of the figure have the same shape but have different representations; that is to say, shape is not effected by its parameterization nor representation.

## A.8 PROOF DETAILS

We include details to the proof of Theorem 4.3. First we show the bound on the upper bound on the privacy loss random variable.

$$
\begin{aligned}
P(PL \geq \epsilon) &= P\left(\log\left[\frac{\exp\{-\frac{1}{2\sigma}(\|h(D)\|_{\mathcal{H}}^2 - 2T_D(x))\}}{\exp\{-\frac{1}{2\sigma}(\|h(D')\|_{\mathcal{H}}^2 - 2T_{D'}(x))\}}\right] \geq \epsilon\right) \\
&= P\left(\log\left[\exp\{-\frac{1}{2\sigma}\left(\|h(D)\|_{\mathcal{H}}^2 - \|h(D')\|_{\mathcal{H}}^2 - 2(T_D - T_{D'})(x)\}\right]\right] \geq \epsilon\right) \\
&= P\left(-\frac{1}{2\sigma^2}(\|h(D)\|_{\mathcal{H}}^2 - \|h(D')\|_{\mathcal{H}}^2 - 2(T_D - T_{D'})(x)) \geq \epsilon\right)
\end{aligned}
$$

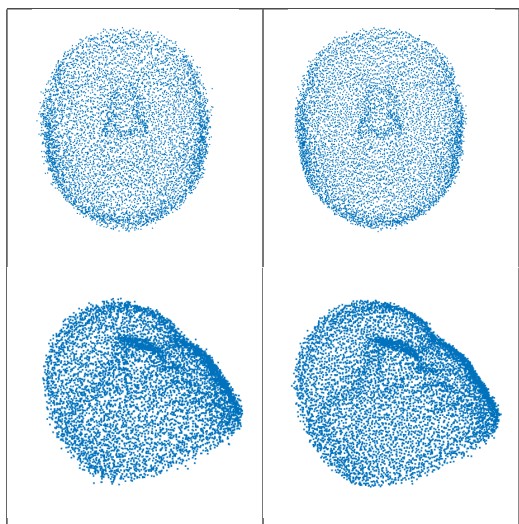

Figure 8: Left: A private face sanitized with Gaussian noise point-wise with total $\mu_T = 2$. Right: A private face sanitized with Gaussian noise point-wise with total $\mu_T = 3$.

Note that we have $\|h(D')\|_{\mathcal{H}} - \|h(D)\|_{\mathcal{H}} = \|h(D) - h(D')\|_{\mathcal{H}} - 2\langle h(D) - h(D'), h(D)\rangle_{\mathcal{H}}$ and $\langle h(D) - h(D'), h(D)\rangle_{\mathcal{H}} = (T_D - T_{D'})(h(D))$. Thus it follows that,

$$P\left(-\frac{1}{2\sigma^2}(\|h(D)\|_{\mathcal{H}}^2 - \|h(D')\|_{\mathcal{H}}^2 - 2(T_D - T_{D'})(x)) \geq \epsilon\right)$$

$$= P\left(-\frac{1}{2\sigma^2}\left(-\|h(D) - h(D')\|_{\mathcal{H}}^2 - 2(T_D - T_{D'})(x - h(D))\right) \geq \epsilon\right)$$

$$\leq P\left(-\frac{1}{2\sigma^2}\left(-\Delta^2 - 2(T_D - T_{D'})(x - h(D))\right) \geq \epsilon\right)$$

$$= P\left((T_D - T_{D'})(x - h(D)) \geq \sigma^2\epsilon - \frac{\Delta^2}{2}\right)$$

$$= P\left(\sigma\Delta Z \geq \sigma^2\epsilon - \frac{\Delta^2}{2}\right)$$

$$= P(Z \geq \frac{\sigma\epsilon}{\Delta} - \frac{\Delta}{2\sigma}) = \Phi\left(-\frac{\sigma\epsilon}{\Delta} + \frac{\Delta}{2\sigma}\right)$$

This is the upper bound we needed of the privacy loss random variable. We similar need a lower bound on the privacy loss random variable. The steps are very similar, we have that,

$$P(PL \leq \epsilon) = P\left(\log\left[\frac{\exp\{-\frac{1}{2\sigma}(\|h(D)\|_{\mathcal{H}}^2 - 2T_D(x))\}}{\exp\{-\frac{1}{2\sigma}(\|h(D')\|_{\mathcal{H}}^2 - 2T_{D'}(x))\}}\right] \leq \epsilon\right)$$

$$= P\left(Z \leq \frac{\sigma\epsilon}{\Delta} - \frac{\Delta}{2\sigma}\right) = \Phi\left(\frac{\sigma\epsilon}{\Delta} - \frac{\Delta}{2\sigma}\right).$$

This completes the proof.

## A.9 ADDITIONAL RESULTS

In 5.1 we describe how to privatize a point-wise average face. Figure 8 displays a comparison of two private faces with $\mu_T = 2$ and $\mu_T = 3$, each from two angles. The two faces look fairly similar, but this is an artifact of the noise making the face rough and fuzzy. The face on the right column has a smoother outline as compared to the face on the left. In Table 1 we see that the MSE of the face on the right is nearly half of that of the face on the left.

In the right panel of Figure 3 we show the coordinate curves of one face radial curve for one face; In Figure 9 we show the coordinate curves of one face radial curve for one hundred faces. We see

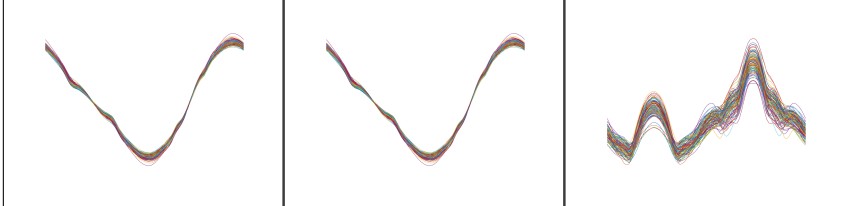

Figure 9: The $x$, $y$, and $z$ coordinate curves, from left to right, of one face radial curve for one hundred individuals.

in Figure 9 that the curves (and hence the features) are aligned across individuals; this empirically shows the alignment and registration achieved by the methodology in Section 2.2.

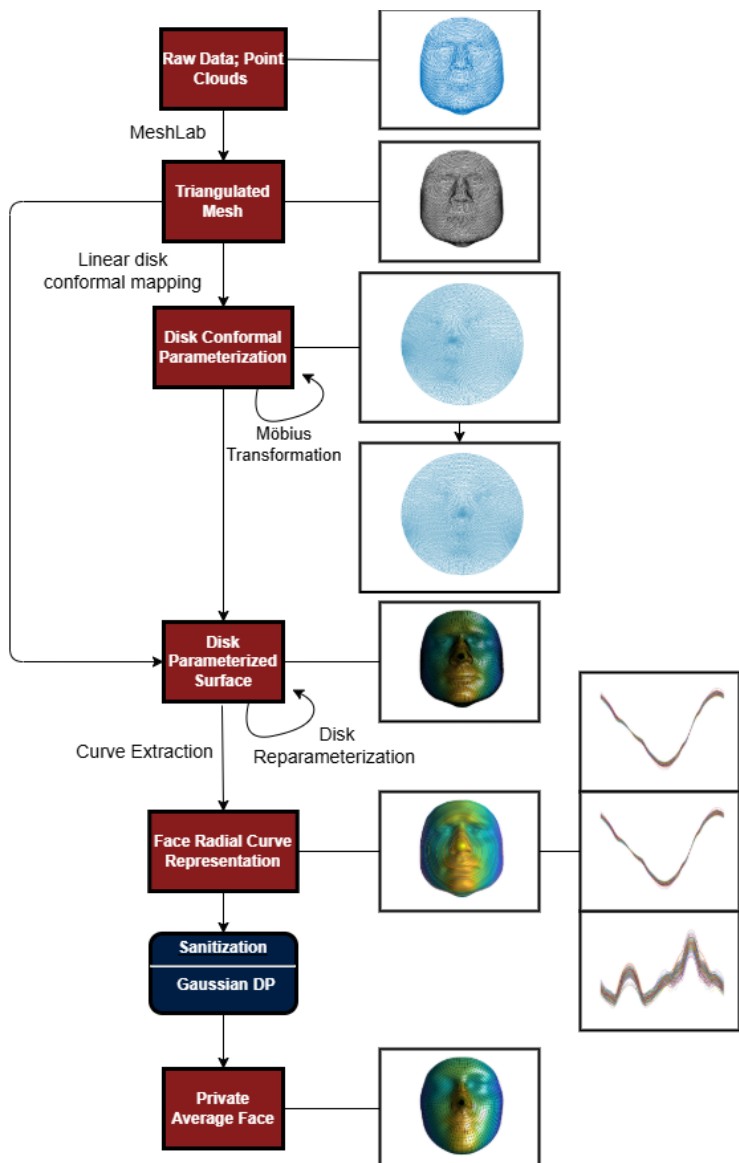

Figure 10: A pipeline diagram of our methodology. We generate disk parameterized surfaces which we register and align to a template face. We extract the face radial curves from the disk parametrization and produce a sanitized mean via our GDP functional data mechanism.

