# OpenReview forum: "Gaussian Differentially Private Human Faces Under a Face Radial Curve Representation"
_ICLR.cc/2025/Conference — ICLR 2025 Poster_

### Official Review · Reviewer_y1vv · 2024-11-04

**Soundness:** 3
**Presentation:** 2
**Contribution:** 3
**Rating:** 6
**Confidence:** 4

**Summary:**

The paper introduces a new representation termed "face radical curves" for a set of 3D faces. It then utilizes the Gaussian Differential Privacy (DP) framework on this representation to create a private average face. The empirical and quantitative results from the experiments show that this method not only maintains the shape of the average face but also introduces less noise compared to traditional methods for the same privacy budget.

**Strengths:**

(1) This paper proposes a novel privacy-preserving representation method for 3D faces. According to the authors, this approach is also applicable to other disk-like surfaces.
(2) The paper presents the mean squared error between private estimates and the point-wise mean, showing that this method outperforms the conventional point-wise method.

**Weaknesses:**

（1) While the article effectively demonstrates the privacy capabilities of the representation through mean squared error metrics for 3D faces, it does not address the usability of the representation for practical applications, such as age estimation and expression analysis of faces. This aspect of usability verification is missing from the paper.
（2）The authors suggest that the method described in the paper could be applicable to domains beyond 3D facial data. It would strengthen the paper if the authors could include experimental results demonstrating this method's effectiveness in other applications as well.

**Questions:**

The paper needs to provide additional evidence to substantiate the usability of the proposed representation.

---

> ### Author Response · Authors · 2024-11-20
>
> We thank the reviewer for their summary of our paper.
>
> Weakness (1) We see the importance of the problems of expression analysis, age estimation, and other related problems in facial recognition, however the emphasis and main goal of this paper is the privacy of the faces, as the reviewer stated. We will expand on the discussion about the applicability of our GDP FDA framework in general settings and add citations on these independent FDA interests. This will emphasize the usability of one of our contributions for privacy in a larger context.
>
>
> Weakness (2) We include most of the response to this in our response is in Weakness(1). We agree that adding another example of disk parameterized surfaces would strengthen the paper; in terms of privacy, human faces seemed like the most relevant example. While one could use this for any disk parameterized surfaces, our face data exists on many datasets and this method being applicable in those settings seem quite pertinent. We will further expand on this but also believe adding a more classical FDA example will strengthen the applicability.

---

### Official Review · Reviewer_rDcW · 2024-11-06

**Soundness:** 2
**Presentation:** 2
**Contribution:** 2
**Rating:** 3
**Confidence:** 2

**Summary:**

This paper proposes to extend  existing approximate Differentially Private (DP) Functional Data analysis (FDA) tools into the Gaussian DP framework. These idea have been applied to the protection of 3D face data, where each face is represented by a collection of a set of curves.

**Strengths:**

The idea of using Functional Data Analysis  Gaussian  mechanism  Differentially Private to protect 3D face is interesting.

**Weaknesses:**

- The writing of this paper should be improved,  it does not facilitate the understanding of the contributions in this paper.
- The approximation of 3D face by a set of radial curves has been published ins several papers related to 3D face recognition.  Hassen (1) Drira, Boulbaba Ben Amor, Mohamed Daoudi, and Anuj Srivastava. Pose and expression invariant
3d face recognition using elastic radial curves. In British machine vision conference, pp.
1–11, 2010. (2) Chafik Samir, Anuj Srivastava, and Mohamed Daoudi. Three-dimensional face recognition using
shapes of facial curves. IEEE Transactions on Pattern Analysis and Machine Intelligence, 28(11):
1858–1863, 2006.
- The contributions of this paper are not clear. Does the contributions of this paper concern the extension of GDP to functional data analysis, or the application to 3D face protection?
- The authors talk about functional data analysis which is a very general term, they should be more precise and discuss the representation of 3D faces by functions?
- In Appendix B.3 refers to the SRNF  representation of surfaces but how this representation is used in this paper ?

**Questions:**

Does the contributions of this paper concern the extension of GDP to functional data analysis, or the application to 3D face protection?
In Appendix B.3 refers to the SRNF  representation of surfaces but how this representation is used in this paper ?

---

> ### Author Response · Authors · 2024-11-20
>
> In response to weaknesses (2) and (5): We are familiar with the mentioned citations and both were cited in our paper. Our representation, while similar, is not entirely the same. The cited works generate the curve/function representation on each face independently. We, however, use the SRNF framework to register all curves/functions across all faces simultaneously. The cited work, for instance, will generate the ith curve of a face as all points distance $r$ from the tip of the nose. This restriction is limiting as features such as eyes are not the same distance from the tip of the nose among all individuals. We have no such restriction. Our use of the SRNF framework forces a registration among all faces and this distinction is the difference and strength in our representation.
>
> In response to weakness (3): We see both mentioned items as contributions. (a) The extension of GDP to FDA is novel; this contribution can be further explored and we will add an example of DP FDA comparisons with classical FDA examples such as the Berkeley growth data. (b) Differential privacy for 3D faces is novel; we agree that ``privacy" for 3D faces has been considered but not \textit{differential} privacy. (c) the representation we propose is novel, albeit incremental in comparison to the previous cited works. In conjunction these three contributions answer a larger question of privacy for human 3D faces.
>
> In response to weakness (4): We speak of FDA in general terms as our contribution in GDP FDA is not limited to faces. Our GDP FDA method can be applied to any set of functions.
> Each face is represented as a set of functions $\{f_i\}$ with each $f_i:\mathbb{S}^1\rightarrow \mathbb{R}^3$, that is each function is parameterized by the unit disk.
>
> We thank the reviewer for their suggestions but note that we have addressed many of these weakness in the original submission. We will however add further discussion on these topics and an example of GDP FDA for functions outside of the scope of privacy for faces upon a resubmission.

---

### Official Review · Reviewer_nNYC · 2024-11-08

**Soundness:** 3
**Presentation:** 1
**Contribution:** 2
**Rating:** 6
**Confidence:** 2

**Summary:**

This paper introduces a method for creating Gaussian Differentially Private (GDP) representations of 3D human faces using a novel face radial curve representation. The proposed approach aims to address privacy concerns associated with sharing 3D facial data by employing differential privacy (DP) mechanisms. This method leverages statistical shape analysis to represent faces in a disk-parameterized structure, ensuring minimal noise addition while preserving the facial structure. The approach is specifically designed for applications that involve facial data but could extend to other disk-like surface data. The paper also presents experimental results demonstrating that the method injects less noise compared to point-wise differential privacy approaches while effectively preserving facial features.

**Strengths:**

1. The paper presents a novel approach to 3D face representation for privacy preservation using face radial curves, which is innovative and highly relevant to privacy concerns in biometric data.

2. The GDP mechanism integrated into this method ensures that facial features are preserved with minimal noise injection, enhancing privacy while maintaining utility.

3. The manuscript includes thorough experiments comparing the proposed method with traditional point-wise differential privacy techniques, demonstrating superior noise reduction and structure preservation.

**Weaknesses:**

1. **Manuscript Organization**: The paper would benefit from a more structured organization aligned with the standard conference paper layout in AI. The current structure is challenging to follow, and the clarity could be enhanced by including diagrams that illustrate the overall pipeline of the proposed method. Guiding sentences that introduce and connect sections and paragraphs would also help readers navigate the content more effectively.

2. **Assumptions on Data Structure**: The proposed approach assumes a genus-0 surface with no missing data points, which may not be realistic in practical settings. This limitation necessitates additional pre-processing for noisy or incomplete data. Expanding the discussion on how the method could handle such cases would improve its practical applicability.

3. **Limited Dataset and Generalizability**: The evaluation is conducted on a limited dataset without a detailed description of its statistics, raising concerns about the method’s generalizability. While the authors suggest that this approach could extend to applications such as terrain models, no experiments support this claim. More comprehensive testing, including diverse or noisy datasets, is recommended to validate the method’s broader applicability.

4. **Computational Efficiency and Benchmarking**: Although the algorithm's methodology is well-described, the computational cost is not evaluated, which limits the ability to assess its practical performance. Additionally, the benchmark methods used for comparison are not clearly described, which could be clarified to provide a more comprehensive context for evaluating the proposed approach.

**Questions:**

Please refer to the weaknesses.

---

> ### Author Response · Authors · 2024-11-20
>
> We thank the reviewer for the thorough summary.
>
> In regards to weakness (1), Diagrams aren't typical in the differential privacy literature. Guiding sentences and a diagram are small, reasonable asks, so we will add these upon a resubmission.
>
> Weakness (2): We will expand on this discussion. We envisioned settings such as anthropologic studies where the data is collected in a uniform, controlled environment such as in the cited Sero et al. paper. However, we agree it is not always the case of having perfect data.
>
> In the case of non perfect data we have two cases (a) Noisy data: this would only pose an issue in the processing step as triangulated meshes are generally not robust to noise. Although we did not explore this avenue, MeshLab does offer smoothing which would circumvent this issue. Further, our GDP functional mechanism incorporates smoothing, so noisy parameterized faces should pose no issue. (b) Missing and incomplete data: this is a case which our method cannot immediately handle. We will expand on the discussion and mention how facial recognition handle the problem of missing data such as in Dagli et al. (2011). This would however only add another pre processing step to our method. Generally these missing data problems for faces use another dataset (or the clean data) to infer the missing components or perhaps a template face. This problem is indeed interesting but also not entirely in the scope of our method. Partial matching of curves for shape analysis has been explored for instance in Bryner \& Srivastava (2021) as well as for face surfaces as in Perakis et al. (2009) which can be interesting avenues for future works. We thank the reviewer for mentioning this weakness however emphasize data imputation in the case of missing data for faces is an important topic which can be handled independently as it is not necessarily an easy task. We will expand on these points in our discussion and mention our data is very clean compared to some real world scenarios.
>
> Weakness (3) Part of this response is contained in our response to Weakness (2). We do suggest our method can be applied to terrain data and any disk parameterized surfaces, in terms of privacy though faces seem to be the most relevant. In our paper we focused on privacy for sets of functions which constituted faces but our GDP FDA contribution is general to any set of functions. We will expand on the discussion about the applicability of our GDP FDA framework in general settings and add citations on these independent FDA interests.
>
>
> Weakness (4) We do mention some computational times in the Supplemental material section ``Experiments Compute Resources." We did not investigate the computational time of the shape analysis pre processing as much of it was explored in Jermyn et al. (2017). Computing the mean and sanitizing it using our method requires a relatively trivial amount of time (less than 3 minutes on a desktop).
>
> We will add more details on the benchmark method but also note that previous to this paper there have been no other works in the \textit{differential} privacy setting for 3D faces. In short, the we add noise to the point-wise mean by adding noise to each coordinate of each point. For comparison to our method, we take the budget from the GDP FDA method and split it evenly across all points to have the same total budget. We will add a table for the computational cost for all the processing and the benchmark method.
>
> The reviewer has pointed out weaknesses for the paper which we believe are important issues which will be discussed upon a resubmission. Some issues however are whole research areas in and of themselves; these issues of course should and will be mentioned.

---

> > ### Comment · Reviewer_nNYC · 2024-11-26
> > **Improving the Ranking to Slightly Above the Threshold**
> >
> > I appreciate the authors' detailed feedback in addressing my concerns. As an emergency reviewer, I must acknowledge that I had relatively limited time to thoroughly review this paper and that I am not an expert in the specific topic this study addresses. After considering the rebuttal and the discussions between the authors and other reviewers, I have decided to increase my score to 6, slightly above the acceptance threshold. In my view, the primary limitation of this paper lies in its presentation, which does not follow the standard flow of AI papers, making it somewhat challenging to follow. However, from a technical perspective, I recognize the contribution of this work and believe it may offer valuable insights for researchers exploring related topics. That said, as reflected in my confidence score, my lack of expertise in this area means my evaluation may not fully capture the paper's value, which ACs and Senior ACs need to consider when making the final decision.

---

### Meta-Review · Area_Chair_GNxk · 2024-12-17

**Metareview:**

The paper introduces a Gaussian Differentially Private (GDP) method for 3D facial data using a novel radial curve-based representation to preserve privacy while maintaining structural fidelity. This paper proposes an innovative approach to integrating GDP with facial representations, and experimental validation showing superior performance in maintaining facial features. Before the rebuttal, the reviewers raise questions about unclear manuscript organization, limited dataset generalizability, and missing evidence of applicability to broader use cases. During the rebuttal, the authors have partially solved the questions. Thus I recommend accepting this paper.

**Additional Comments On Reviewer Discussion:**

Reviewers raised concerns about the paper's presentation, assumptions on perfect data, applicability to broader domains, computational efficiency, and contributions' clarity. The authors responded by committing to improve presentation with diagrams and guiding sentences, addressing noisy and missing data challenges, and expanding discussions on applicability to general FDA settings. They clarified their contributions, including SRNF-based registration and GDP extension to FDA, and provided details on computational resources and benchmarks. While some weaknesses were acknowledged as broader research areas, the authors' detailed responses demonstrated a strong technical foundation. The revisions addressed key concerns, influencing the final decision positively.

---

### Decision · Program_Chairs · 2025-01-22

Accept (Poster)